# Comparative Genomic Analyses of *Colletotrichum lindemuthianum* Pathotypes with Different Virulence Levels and Lifestyles

**DOI:** 10.3390/jof10090651

**Published:** 2024-09-13

**Authors:** Ma. Irene Morelos-Martínez, Horacio Cano-Camacho, Karla Morelia Díaz-Tapia, June Simpson, Everardo López-Romero, María Guadalupe Zavala-Páramo

**Affiliations:** 1Centro Multidisciplinario de Estudios en Biotecnología, FMVZ, Universidad Michoacana de San Nicolás de Hidalgo, Km 9.5 Carretera Morelia-Zinapécuaro, Posta Veterinaria, Morelia 58000, Michoacán, Mexico; irene.morelos@umich.mx (M.I.M.-M.); hcano1gz1@mac.com (H.C.-C.); morelia.diaz@umich.mx (K.M.D.-T.); 2Centro de Investigación y Estudios Avanzados del Instituto Politécnico Nacional, Unidad Irapuato, Km 9.6 Libramiento Norte Carretera Irapuato-León, Irapuato 36821, Guanajuato, Mexico; june.simpson@cinvestav.mx; 3Departamento de Biología, División de Ciencias Naturales y Exactas, Universidad de Guanajuato, Noria Alta SN, Guanajuato 36030, Guanajuato, Mexico; everlope@ugto.mx

**Keywords:** genomic, pathotypes, intraspecific diversity, transposable elements, TFomes, proteases, effectors, virulence genes, CAZymes

## Abstract

*Colletotrichum lindemuthianum* is the most frequent pathogenic fungus of the common bean *Phaseolus vulgaris*. This filamentous fungus employs a hemibiotrophic nutrition/infection strategy, which is characteristic of many *Colletotrichum* species. Due to host–pathogen coevolution, *C. lindemuthianum* includes pathotypes with a diversity of virulence against differential common bean varieties. In this study, we performed comparative genomic analyses on three pathotypes with different virulence levels and a non-pathogenic pathotype, isolated from different geographical areas in Mexico. Our results revealed large genomes with high transposable element contents that have undergone expansions, generating intraspecific diversity. All the pathotypes exhibited a similar number of clusters of orthologous genes (COGs) and Gene Ontology (GO) terms. TFomes contain families that are typical in fungal genomes; however, they show different contents between pathotypes, mainly in transcription factors with the fungal-specific TF and Zn2Cys6 domains. Peptidase families mainly contain abundant serine peptidases, metallopeptidases, and cysteine peptidases. In the secretomes, the number of genes differed between the pathotypes, with a high percentage of candidate effectors. Both the virulence gene and CAZyme gene content for each pathotype was abundant and diverse, and the latter was enriched in hemicellulolytic enzymes. We provide new insights into the nature of intraspecific diversity among *C. lindemuthianum* pathotypes and the origin of their ability to rapidly adapt to genetic changes in its host and environmental conditions.

## 1. Introduction

The ascomycete *Colletotrichum lindemuthianum* causes anthracnose disease in common bean plants (*Phaseolus vulgaris*). Worldwide, anthracnose is one of the main diseases affecting different common bean varieties, causing losses of up to 100% [1,2,3,4,5]. This species has a hemibiotrophic lifestyle or a nutrition/infection strategy, consisting of a biotrophic phase when it infects and feeds on the host’s living tissues, followed by a necrotrophic phase when it kills the tissues and continues its nutrition [6,7,8,9]. The interaction of *C. lindemuthianum* with varieties of *P. vulgaris* worldwide has given rise to a host–pathogen coevolution process, generating fungal races or pathotypes with different levels of virulence against differential bean varieties [10,11,12,13]. Pathotype diversity is assessed based on the interaction of fungal isolates following inoculation on a globally accepted binary system of 12 differential bean cultivars of Mesoamerican and Andean origin [14,15]. The virulence index (VI) of each pathotype is calculated using the number of susceptible bean cultivars × 100/the total number of differential cultivars [13]. Recently, 298 races or pathotypes have been reported distributed in 29 countries [13]. *Colletotrichum lindemuthianum* pathotypes from Latin America have developed an adaptation to local hosts and two genetic groups associated with the genetic groups of *P. vulgaris* from the Mesoamerican, northern, and southern Andean diversification centers [10,11,16]. Furthermore, worldwide, several isolates have been classified as pathotype 0 (P0) because they failed to infect any differential bean cultivar from the binary system, suggesting a saprophytic lifestyle preference [10,12,13]. This behavior indicates the high genomic plasticity of pathogenic fungi, which allows them to rapidly adapt to the host range, host evolution, and environmental changes [17,18].

In recent decades, genomic and transcriptomic studies on fungal species of the genus *Colletotrichum* have increased considerably, revealing interesting aspects of plant–pathogen interaction. For example, they can contract or expand gene families involved in pathogenicity in relation to the reduction or expansion of the host range during evolution [19]. Fungi with a hemibiotrophic and necrotrophic nutrition/infection strategy have also been found to have a broader repertoire of genes for Carbohydrate Active Enzymes (CAZymes) compared to biotrophic or saprophytic fungi [20,21]. This genomic plasticity is mainly attributed to point mutations induced by repeats (RIPs) and/or transposable elements (TEs), which are located in regions close to genes involved in pathogenicity, such as genes encoding CAZymes and proteins involved in the synthesis of secondary metabolites and proteases [22,23]. Recently, intraspecific comparative genomic analyses of *Colletotrichum* species strains with different virulence levels have shown genome rearrangements and differences in the candidate effector gene content [24,25]. Furthermore, it has been found that genomes of filamentous plant pathogens have a dualistic architecture according to the two-speed model of evolution, i.e., a core genome with housekeeping genes and a lifestyle-adapting genome with TEs and effector genes [26,27,28].

Previously, 64 *C. lindemuthianum* pathotypes were identified in several isolates collected from different geographic regions of Mexico, in which different bean cultivars (bush- and climbing-type) are grown under different conditions (in large areas on a commercial scale or in small plots mixed with maize) [10,12,13,29]. Using Amplified Fragment Length Polymorphism (AFLP), molecular analysis was performed on 59 isolates belonging to 10 pathotypes (included P0). They showed a high genetic diversity, detecting two genetic groups associated with northern Mexico and two others with central [10]. Despite the evidence of virulence and lifestyle diversity in *C. lindemuthianum*, there are no comparative genomic studies of their pathotypes. Intraspecific analyses will provide insight into the origin of their ability to rapidly adapt to genetic changes in their host and environmental conditions.

In this study we describe the genome sequences, assembly, and functional annotation of three *C. lindemuthianum* pathotypes with different virulence levels (P1088, P1472, and P2395) and one non-pathogenic pathotype (P0). We performed a comparative genomic analysis that included the determination of TE content and associated divergence dynamics, an estimation of the number of genes and proteins, InterPro and PFAM protein domains, a phylogenomic analysis, and the determination of clusters of orthologous genes (COGs). Gene Ontology (GO) terms were determined and data for specific genes/proteins of interest was generated including transcription factors, peptidases, proteins with signal peptide, proteins with transmembrane domain, secretomes, effectors, virulence genes, and CAZymes.

## 2. Materials and Methods

### 2.1. Pathotypes and Culture Conditions

*Colletotrichum lindemuthianum* pathotypes P0 (VI = 0.0) and P1472 (VI = 33.3) were collected from central Mexico (Michoacan and Jalisco) from climbing-type cultivars grown in small plots in combination with maize. In addition, P1088 (VI = 16.6) was collected from northern Mexico (Durango) from commercial-scale bean-producing areas [10] (For details see Díaz-Tapia et al. [30]). Pathotype P2395 (VI = 58.3) was collected from central Mexico (Guanajuato) from beans grown on a small scale and was kindly provided by Dra. Brenda Z. Guerrero-Aguilar, Dr. José L. Pons-Hernández, and Dr. Raul Rodríguez-Guerra from the Centro de Investigación Regional del Centro-INIFAP, Celaya, Mexico (For details see Díaz-Tapia et al. 2024 [30]). The fungi were maintained on potato dextrose agar (PDA) prepared according to the method of French and Hebert [31]. For DNA extraction, the mycelium of each pathotype grown on PDA plates was inoculated into 125 mL-Erlenmeyer flasks containing 50 mL of potato dextrose (PD) medium [31] and shaken at 150 rpm and 18 °C for seven days. Subsequently, mycelia were collected by filtration using a 3 mm Whatman filter paper; washed with sterile, deionized water; and stored at −80 °C until use.

### 2.2. Genomic DNA Preparation, Library Construction, and Sequencing

Genomic DNA was isolated from seven-day-old mycelium of each pathotype grown in PD medium, using the DNeasy^®^ Plant Mini Kit (QIAGEN, Hilden, Germany). The DNA quality and concentration were verified using agarose gel electrophoresis and a Nanodrop (Thermo Scientific, Waltham, MA, USA). Library construction and whole genome sequencing were performed by Psomagen Inc. (Psomagen Inc., Rockville, MD, USA). DNA quality control was performed with the Genomic DNA ScreenTape assay on a TapeStation (Agilent Santa Clara, Santa Clara, CA, USA). Library construction was carried out with the Nextera DNA flex system (Illumina Inc., San Diego, CA, USA). Library quality and concentration control were assessed using a DNA QC-Picogreen and D5000 Screen Tape on a TapeStation (Agilent Santa Clara, Santa Clara, CA, USA). Next-generation sequencing of the four libraries was performed on the Illumina NovaSeq 6000 S4 150PE platform (Illumina Inc. San Diego, CA, USA). Bioinformatic processing was performed by BioScience App Inc. (BioScience App Inc., Lima, Peru).

### 2.3. De Novo Genome Assembly, Gene Prediction, and Transposable Element Annotation

The FastQC v0.11.5 software [32] was used to remove low-quality reads, and Trimmomatic v0.36 [33] was used for raw data filtering. Clean reads were de novo assembled using SPAdes v3.15.4 [34] with default parameters, and the quality of the assemblies and the number of predicted genes were assessed using QUAST v5.2 [35]. The annotation of repetitive and transposable elements (TEs) for the fungi and divergent dynamics were performed using Earl Grey v4.03 [36]. The whole-genome sequencing and assembly datasets from this study have been submitted to the National Center for Biotechnology Information (NCBI) database under the BioProject accession number PRJNA1147206.

### 2.4. Genome Functional Annotation

Functional annotation was focused on identifying transcription sites, conserved protein domains, tRNAs, genes, and their biological function. Moreover, phylogenomic relationships using Maximum likelihood (ML) and single-copy orthologous genes were assessed using Funannotate v1.8.15 [37] and Gene prediction and phylogenomics BUSCO v5.4.7 [38]. The annotations obtained in GBK format were used for comparison at the level of genomic products between pathotypes, which were plotted in a Venn diagram constructed using the Intervene 1.0.0 tool [39]. We used the following databases: BUSCO data sets v5.4.7 [38], Functional analysis of proteins InterPro v97.0 [40], Protein Families PFAM 36.0 with algorithms blastx and blastp, and alignments using Diamond v2.1.8 [41], the clusters of orthologous genes (COGs) database (NCBI), Gene Ontology (GO) [42,43,44], Fungal Transcription Factors (TFs) [45], and Peptidase database MEROPS v12.5 [46]. The prediction of the protein transmembrane helices was performed using the TMHMM v2.0 software. To identify genes encoding secreted proteins, those proteins predicted by Funannotate were analyzed for signal peptide and transmembrane domain identification using the SignalP v4.1 [47] and TMHMM v2.0 (https://services.healthtech.dtu.dk/services/TMHMM-2.0/, accessed on 7 November 2023) tools. Common annotations with transmembrane domains were excluded to obtain only proteins with a unique signal peptide. Effector prediction was carried out using EffectorP 3.0-fungi [48]. Virulence genes prediction was performed using the Pathogen Host Interactions (PHI-base) database [49], blastn and blastp algorithms. Venn diagrams for comparing virulence genes were constructed using the Intervene 1.0.0 tool [39]. In addition, we used the Carbohydrate Active Enzymes Database (CAZy) (http://www.cazy.org/, accessed on 7 November 2023) [50]. Heatmap visualizations of the TFs, peptidase families, and CAZymes were constructed with the GraphPad software v10.0.3 (GraphPad PRISM).

## 3. Results and Discussion

### 3.1. Genome Assembly, Gene Prediction, and Transposable Elements Annotation

Genome sequencing generated 32,424,538 reads for P0, 26,793,550 for P1088, 32,183,896 for P1472, and 31,500,326 for P2395. The total number of bases (bp) were 4.90 Gb for P0, 4.0 for P1088, 4.86 for P1472, and 4.76 for P2395. After quality control, raw data filtering, and assembly, the four whole genomes exhibited a range of sizes from 98.1 to 101.8 Mb and were assembled into 24,410 to 29,513 scaffolds, with a N50 from 28.42 to 33.05 Mb, a GC content from 38.7 to 38.9%, and 11,859 to 12,225 predicted genes (Table 1).

A comparison of the Mexican *C. lindemuthianum* pathotype genome assemblies with data for two strains from Brazil available in NCBI showed some differences in genome sizes and GC content (Table 1). The pathotypes or strains S83 and S89 of *C. lindemuthianum* were reported as strains [51], have not been identified by confrontation with the 12 differential bean cultivars of Mesoamerican and Andean origin [14,15], and are therefore referred to as strains. Although P2395 exhibited the largest genome size, the number of predicted genes was higher for P1088 than for the other pathotypes. Consistent with the results of previous *Colletotrichum* genome analysis [52,53], our results showed that the genome sizes of the Mexican *C. lindemuthianum* pathotypes are among the largest in the genus. However, the number of predicted genes is similar to or lower than that of other species with smaller genome sizes; for example, *C. fructicola* Nara gc5 has 17,388 genes (Table 1). These differences in genome sizes and gene content may be due to the presence of repetitive and transposable elements. Transposable elements have previously been reported to generate genomic diversity, intraspecific variation, and the evolution of adaptive behavior in phytopathogenic fungi [28,54,55].

Therefore, transposable element (TE) annotation was performed using Earl Grey v4.03 [36] on the four genome assemblies from our study and the strain S89 genome assembly from Brazil (Table 1). The annotated sequences of TEs in the genomes of Mexican pathotypes constituted 48.78% for P0, 48.3% for P1088, 48.8% for 1472, and 48.93% for P2395, whereas for the Brazilian strain S89, the percentage was higher (54.39%) (Figure 1).

The percentages of TEs in *C. lindemuthianum* pathotype genomes were higher than *C. orbiculare* (44.88%); however, the percentage for strain S89 was similar to *C. trifoli* (54.67%) [53,56,57]. Furthermore, the percentages of TEs in these three species were higher than in other species of the genus, which ranged from 4.31% in *C. scovillei* to 27.23% in *C. sublineola* [53,56,57]. We found intraspecific diversity across the *C. lindemuthianum* pathotypes and strain S89; however, TEs were more abundant in the Mexican pathotypes, ranging from 70,899 in P1472 to 78,182 in P1088. The Brazilian strain S89 exhibited a lower TE number of 30,989 (Table 2). However, it is interesting to note that two pathotypes showed a higher TE content (P1088 = 78,182 and P2395 = 77,950) than the other two (P0 = 72,878 and P1472 = 70,899). On the other hand, the genome of strain S89 had a lower number of TEs and contained longer sequences than the Mexican pathotypes, although its percentage coverage was similar to that of the Mexican pathotypes (Figure 1 and Table 2). Furthermore, the Mexican pathotypes presented a high TEs content diversity (Table 2). These results suggest that the Mexican pathotypes went through an evolutionary history distinct to that of the Brazilian strain.

A high abundance of unclassified TE annotations was found in all pathotypes; however, the Mexican pathotypes exhibited a greater and more variable abundance (ranging between 25,110 and 31,524) than strain S89 (14,133) (Table 2). The high level of unclassified TEs in *C. lindemuthianum* pathotypes is commonly observed in other *Colletotrichum* species, indicating a high abundance of unknown TEs in the genus [56]. Of the classified TEs, four families belong to class I and two to class II. Retrotransposons of the Long Terminal Repeat (LTRs) family (class I) were the most abundant, with a range from 36,401 to 47,786 over the Mexican pathotypes, while the Brazilian strain S89 had 14,814. These results are consistent with those reported previously for filamentous fungal genomes, where LTRs are the most common and abundant TEs, mainly in the Gypsy and Copia superfamilies [56,57,58,59]. In addition, a high TE content from the DNA family (class II) was found, of which the pathotype P0 exhibited the highest abundance (3632), while the strain S89 exhibited the lowest (827) (Table 2). The PENELOPE family was detected in all five genomes ranging from 108 to 240 TEs, while the Long Interspersed Nuclear Elements (LINEs) family was only found in P0 (140) and P1088 (96). Moreover, the Short Interspersed Nuclear Elements (SINEs) family was only present in P1472 (18), P2395 (17), and S89 (18) (Table 2). The TE Rolling Circle family (class II) was only present in P0 (36), P1472 (15), P2395 (17), and S89 (18). Other TEs identified in all pathotypes, included simple repeats, microsatellites, and RNAs (Table 2).

According to reports on fungal phytopathogens [23,28,54,55], the content and diversity of intraspecific TEs, such as those found in *C. lindemuthianum*, may influence genome plasticity and evolution, lifestyle, pathogenicity, and adaptation to the genetic variations of the host. TEs can affect the expression of genes involved in virulence in strains or pathotypes of the same fungal species, and the presence or absence of certain TEs can vary, generating new races or biotypes. An example of this can be found in race two of *Fusarium oxysporum* f.sp. *lycopersici*, in which a new biotype emerged due to a mutation derived from a *Hormin* transposon (of the family hAT or DNA transposons) in the avirulence gene AVR1 [60]. Recently, a study of different lineages (pathotypes) of *Magnaporthe oryzae* revealed intraspecific diversity in the TE content, whose expansion dynamics led to the pathotypes that follow different evolutionary histories [55].

We analyzed the divergent dynamics using the Earl Grey tool with the calcDevergence FromAlign.pl utility of RepeatMasker, to determine the genetic distance between each identified TE and its respective consensus sequence [36]. Plots indicate patterns of TE activity; for instance, recently active TE copies are assumed to have low levels of genetic distance to their respective family consensus [36]. All Mexican pathotypes and the Brazilian strain of *C. lindemuthianum* exhibited unclassified TEs, LTRs, and DNA TEs with ancient and recent divergence activity as well as low and high levels of genetic distance (Figure 2). In particular, the Mexican pathotypes exhibited LTRs that have experienced ancient and high divergence. Moreover, other LTRs exhibited an ancient and minor divergence, and there was another group with a minor and recent divergence activity. The unclassified TEs in the Mexican pathotypes exhibited a similar but minor divergence pattern with respect to LTRs; however, the P0 pathotype showed a greater divergence activity in comparison to the other pathotypes. The Brazilian strain S89 presented a similar pattern to that of the Mexican pathotypes; however, the K-value 5 suggests a drastic and ancient reduction in divergence activity (Figure 2). The study in *M. oryzae* revealed that large expansions are the predominant cause of lineage-specific (pathotypes) and isolate specific LTR-retrotransposon copy number variation. Furthermore, the effectors were closer to all the expanded TEs, suggesting a pattern of two-speed genome compartmentalization [55].

The presence of repeated elements clustered in hypervariable minichromosomes (from two to six) in addition to large chromosomes (four to nine) in 10 *C. lindemuthianum* strains has been reported previously [61]. Our study revealed TE expansion patterns in the Mexican pathotypes and the Brazilian strain of *C. lindemuthianum*, which is consistent with their location on hypervariable minichromosomes and with a two-speed genome compartmentalization pattern [26,27,28]. In future studies, the assembly of mini and core chromosomes will allow more detailed analyses.

### 3.2. Genome Functional Annotation

The comparison of genomic products using GBK format annotations revealed a high diversity, with each pathotype showing a different number of genomic products. These were highest in P1088 (19,060) and lowest in P0 (16,843), and from these annotations, 12,852 were shared by all pathotypes. In addition, there were other annotations that were only shared between two or three pathotypes (Figure 3).

The number of proteins varied between pathotypes from 11,434 in P2395 to 11,797 in P1088 (Table 3), and they were relatively similar to those predicted for the Brazilian *C. lindemuthianum* strains S83 (11,673) and S89 (11,627) [51] (Table 1).

The number of proteins with at least one orthologous gene ranged from 11,116 in P2395 to 11,368 in P1472, while single-copy orthologous genes were identical for all pathotypes (10,711) (Table 3). The P1088 pathotype, which showed the highest number of genes, also showed the highest number of proteins, tRNAs, and unique proteins. Interestingly, each pathotype exhibited unique proteins, ranging in number from 98 to 468. However, this does not seem to be related to the level of virulence or lifestyle, since P0 and P1472 had the lowest and similar number of unique proteins, while P1088 and P2395 showed the highest number. The InterPro domains of proteins per genome ranged from 12,901 for P0 to 13,049 for P1088 (Table 3 and Appendix A). In addition, the PFAM domains ranged from 12,528 for P0 to 12,716 for P1088 (Table 3 and Appendix A). For the Brazilian strains S83 and S89, the number of PFAM domains reported (8507 and 8479, respectively) was lower than for the Mexican pathotypes [53]. At the interspecific level, the number of PFAM domains reported for *C. sidae* (9104) and *C. spinosum* (9049) [53] was also lower. The phylogenomic analysis carried out in this study using single-copy orthologous genes by means of the BUSCO tool and the ML method showed pathotypes P0 and P1472 in a basal clade and a subclade including P1088 and P2395, suggesting that pathotypes belong to two genetic groups (Figure 4).

Approximately 50% of the annotated single-copy orthologous genes for all pathotypes could be assigned to COGs (Table 3). Pathotype P1088 had a higher number of orthologous genes (5503) than P0 (5439), P1472 (5452), and P2395 (5470) (Figure 5). However, the number of orthologous genes was lower than in other *Colletotrichum* species, e.g., a total of 7306 orthologous genes have been identified as conserved in *C. fructicola*, *C. orbiculare*, *C. gloeosporioides*, *C. sublineola*, and *C. fioriniae* [62].

In general, the COGs of all pathotypes exhibited the same pattern including one with genes of unknown function (135 to P0, 135 to P1088, 130 to 1472, and 138 to P2395); however, the COGs of carbohydrate transport and metabolism (functional category G) were highlighted because they had the highest number of genes (749 to P0, 760 to P1088, 749 to P1472, and 752 to P2395). Considering that this genus uses the cell wall of the host plant as the main nutritional carbon source, CAZyme genes are expected to be the most abundant in the COGs. For example, 572 CAZymes have been reported in *C. graminicola* and 689 in *C. higginsianum* [20]. The COGs of secondary metabolite biosynthesis, transport and catabolism exhibited 348 genes for P0, 355 for P1088, 349 for P1472, and 353 for P2395. The number of these genes was higher than in *C. higginsianum* (103) and in *C. graminicola* (74) [20]. In *Colletotrichum* species, these genes are diverse. Secondary metabolism includes enzymes such as P450 monooxygenases, which are used for antimicrobial detoxification; polyketide synthases (PKS), which are important in appressorium and macrolide synthesis with phytotoxic and antimicrobial functions; terpene synthases (TSs) for antimicrobial triterpenoid and phytotoxic monoterpenoid synthesis; non-ribosomal peptide synthases (NRPS) for phytotoxic and antimicrobial compounds synthesis; and dimethylallyl transferases (DMAT) [19].

The annotation of unique GO terms belonging to at least one of the three GO_ROOT classes exhibited similar patterns across the pathotypes: 1871 genes for P0, 1880 for P1088, 1875 for P1472, and 1873 for P2395 (Appendix A). In detail, the molecular function ranged from 863 to 869, the biological function ranged from 733 to 738, and the cellular component ranged from 271 to 273 [44]. The mapping of GO terms from each pathotype to 127 terms of the GO slim ancestor revealed terms that belong to at least one of the ninety-one GO slim classes [44]. The total number of genes for the GO terms annotated (with their frequencies) revealed slight differences among the pathotypes. In pathotype P0, 6018 genes were identified, whereas for P1088, 6045 genes were identified, and for pathotypes P1472 and P2395, 6029 and 6027 genes were identified respectively (Appendix A). The most abundant GO slim terms were those for catalytic activity (629 to 633 terms) and metabolism (484 to 486 terms), while 10 GO slim classes showed abundances of 256 to 88 terms. Other GO slim classes were less frequent, and some were found in only two or three pathotypes (Appendix A).

The repertoire of transcription factors (TFs) or TFomes identified between the pathotypes revealed 45 families typically found in more than 200 fungal genomes [63]. Each pathotype exhibited a different total number of genes: 395 in P0, 423 in P1088, 394 in P1472, and 403 in P2395 (Appendix A). However, two pathotypes (P1088 and P2395) had a higher and similar number of genes whereas the other two (P0 and P1472) a lower and similar number. As proposed by Shelest [63], 34 families are small (with less than five genes), and 11 families are abundant (with five or more genes) (Figure 6). Two families exhibited the highest gene abundance: the fungal-specific TF domain family (ranging from 127 to 136 genes among pathotypes) and the fungal Zn(2)-Cys(6) binuclear cluster domain family (Zn2Cys6_DnaBD) (ranging from 110 to 131 genes among pathotypes) (Figure 6 and Appendix A).

This is consistent with what has been reported for other fungi, in which the largest class of TFs is “zinc fingers”, comprising several families such as Zn2Cys6_DnaBD, C2H2-type, and GATA [63,64]. The number of Zn2Cys6_DnaBD family genes in *C. lindemuthianum* pathotypes (110 to 131) was reduced in comparison to that reported in an isolate of *C. graminicola* (230 genes) [63]. This family includes TFs such as the xylanolytic activator xlnR, the arabinanolytic activator araR, the mannanolytic activator ManR, the cellulolytic and xylanolytic activator ACEII, galactose catabolism regulator (GAL4), and the nitrogen assimilation regulator nirA [65,66,67,68]. The Zn2Cys6_DnaBD family also possesses TF genes that regulate virulence in phytopathogenic fungi. For example, Mtf4participates in the development of the appressorium in *C. orbiculare*, which is a necessary structure for penetration of the host [69], and CLTA1is a pathogenicity gene and a regulator of the switch between biotrophy and necrotrophy during the infection process in *C. lindemuthianum* [70]. Additionally, the regulator of aflatoxin biosynthesis aflR was identified [71]. In the Zinc finger C2H2-type family, other TFs involved in virulence were also found, such as those involved in melanization (Cmr1) [72], the regulator of pathogenicity and appressorium formation in *C. lindemuthianum* (CLSTE12) [73], the carbon catabolic repressor (CreA) [73,74,75], and the regulator of the Pal-pH pathway involved in responses to ambient pH (PacC) [76]. The Zinc finger GATA-type family includes the AreA/NIT2-like global nitrogen regulator, which is involved in nitrogen assimilation and triggers the expression of pathogenicity genes [67,77]. The bZIP TF 1 family includes the ROS response regulators Yap1 and Atf1 [78,79]. The differences found among the pathotypes regarding the number of genes with the fungal-specific TF domain and with the Zn2Cys6 domain, particularly those with a function in virulence, could be related to their ability to infect the host, since pathotypes P1088 and P2395 exhibited a greater number of genes from these families.

The annotation of peptidases and protease inhibitors using the MEROPS database, showed seven peptidase families from 26 clans and three protease inhibitor families, with a similar total number of genes across the pathotypes ranging from 405 to 409 (Table 4, Figure 7, and Appendix A).

These numbers of peptidases identified were higher than those previously reported for other *Colletotrichum* species, which ranged from 83 genes in *C. graminicola* to 112 in *C. fructicola* [19]. All pathotypes exhibited a high abundance of serine peptidases (182 to 183), metallopeptidases (110 to 114), and cysteine peptidases (62 to 63) (Table 4, Figure 7, and Appendix A). Serine peptidases belonging to nine clans containing 15 families were identified, including peptidases involved in cytoplasmic, nuclear, mitochondrial, lysosomal, vacuolar, and extracellular function, and others with unknown biological functions. Serine peptidase genes of prolyl oligopeptidases and prolyl aminopeptidases were the most abundant, followed by subtilisin genes (Figure 7). Subtilisins are ubiquitous in fungi with a function in nutrient acquisition and are believed to play different roles in adaptation to distinct ecological niches [80]. A genomic comparison of 83 pathogenic and saprophytic fungi showed that the presence of multiple subtilisin-encoding genes is not associated with pathogenicity in some fungi [80]. In contrast, it is proposed that the presence of multiple subtilisins in pathogenic fungi could contribute to higher adaptability, increased host range, and/or survival in diverse ecological habitats outside the host [80].

Metallopeptidases belonging to eight clans containing 33 families were the most diverse. They included peptidases for nuclear, cytoplasmic, vacuolar, mitochondrial, and extracellular (secreted) functions, and others whose biological functions are unknown (Figure 7). Secretion metallopetidases of included a fungalysin gene (family M36), which is a virulence factor involved in the degradation of chitinases in *C. graminicola* and other fungi [81,82]. Moreover, four deuterolysin genes (M35 family) were identified in *M. oryzae*, one of which has a the role of an effector [83] and three to four cytophagalysin genes (M43B family) were identified in *Phytophthora infestans*, described as necessary for virulence [84]. The cysteine peptidases identified belong to five clans containing 18 families, including peptidases for cytoplasmic, nuclear, and lysosomal functions and others with unknown functions (Figure 7). The genes for calpains and caspases, and those involved in ubiquitin release, are found among cysteine peptidases. Threonine peptidases belong to one clan with four families, including proteases with functions in the endoplasmic reticulum, Golgi, proteasome, and metabolism (Figure 7). Aspartic peptidases belong to one clan and one family, and mainly include members of the pepsin family. In addition, the glutamic peptidase corresponds to fungal scytalidoglutamic endopeptidase and mixed peptidase is an aminopeptidase (Figure 7).

Regarding secretome prediction, the number of genes was different between the pathotypes, ranging from 1126 for P0 to 1197 for P1088 (Table 5).

Pathotypes P1088 and P0 exhibited the highest and lowest number of genes encoding secretory proteins, respectively. The number of genes reported in the secretomes of Brazilian *C. lindemuthianum* strains P83 and P89 (1150 and 1136, respectively) [53] were similar to those of the Mexican pathotypes. Furthermore, the number of genes in the *C. lindemuthianum* pathotypes was higher than in the *C. incanum* strain (1002) [62], while it was lower than in *C. orbiculare* (1235) [53] and that of *C. gloeosporioides* Lg1 (2047) [85].

### 3.3. Effectors

Proteins filtered by SignalP were subjected to effector prediction using the EffectorP 3.0-fungi tool, which focuses on fungal sequences in secretomes that distinguish between apoplastic and cytoplasmic effectors [48]. Apoplastic effectors are secreted into the extracellular space (the plant apoplast), where some of them modulate the activity of plant defense enzymes, while others bind to the fungal cell wall. Cytoplasmic effectors are delivered into the plant cytoplasm, where they regulate different molecular and metabolic processes [86]. The total number of predicted candidate effectors (CEs) was highest in P1088 (475) and lowest in P2395 (432), while P0 and P1472 had a similar number (443 and 445, respectively) (Table 6 and Appendix A). However, the CEs of each pathotype accounted for different percentages of their secretome: 39.34% in pathotype P0, 39.68% in P1088, 39.1% in P1472, and 38.02% in P2395. All pathotypes showed a greater number of apoplastic effectors in comparison to cytoplasmic CEs.

The total content of predicted CEs in the four Mexican pathotypes was higher than that found in the Brazilian *C. lindemuthianum* strains S83 (370) [87] and S89 (349) [51]. Recently, a comparative genomic analysis revealed a large and variable content of CEs in Colletotrichum species, ranging from 288 to 608 per genome [87]. Furthermore, an intraspecific comparative genomic analysis of strains with different levels of virulence of *C. scovillei* and *C. lini* showed genome rearrangements and differences in the content of CEs genes [24,25]. In Colletotrichum species, the presence of minichromosomes that are highly enriched in TEs and closely associated with genes encoding putative effectors is common [22,23,56,87,88,89]. In this sense, since the minichromosomes from 10 *C. lindemuthianum* strains exhibit a high proportion of clustered repeated sequences [61], then similar patterns of repeated sequences and TEs must exist in any strain or pathotype of the species. Furthermore, as observed in the genus, CEs are likely to be associated with TEs within minichromosomes. We propose that the high and variable TE content showing expansion patterns, and their association with a high CE content within minichromosomes in *C. lindemuthianum* pathotypes, is consistent with an architecture that is related to the lifestyle-adapting component of the two-speed model of evolution [26,27,28].

The CE repertoire found in *Colletotrichum* species is mainly composed of small cysteine-rich proteins [22,87,90,91,92]. Herein, we do not present a detailed analysis of CEs in *C. lindemuthianum* pathotypes, a subject that will be part of a future study; however, an analysis by de Queiroz et al. [51], performed in *C. lindemuthianum* strains S83 and S89, compared small cysteine-rich proteins with those from other *Colletotrichum* species. They found that 63% of all CEs are cysteine-rich proteins with repetitive sequences. Furthermore, 20 predicted conserved domains for CEs were detected, and an expression analysis of eight CE-encoding genes showed induction during the biotrophic phase of the fungus in bean [51]. In our peptidase annotation results, we identified the metalloprotease gene fungalysin belonging to the MEROPS M36 family (Figure 7), a highly conserved apoplastic effector in fungi whose function is the degradation of chitinases secreted by the host plant [93]. Fungalysin gene expression in *C. graminicola* occurs during the biotrophic phase of infection and also has a role in virulence [81]. In addition, four deuterolysin genes (M35 family) which have effector functions in *M. oryzae* were identified [83].

### 3.4. Virulence Genes

Virulence gene prediction was performed using the plant–pathogen interactions (PHI) database (http://www.phi-base.org/index.jsp, accessed on 7 November 2023). A total of 950 virulence genes were identified in the *C. lindemuthianum* pathotypes, of which 444 were shared (Figure 8).

The virulence gene content for each pathotype was variable, i.e., 652 genes in pathotype P0, 711 in P1088, 649 in P1472, and 682 in P2395. Some genes were shared by only two or three pathotypes and each pathotype exhibited unique virulence genes (Figure 8). In particular, the P2395 pathotype (the most virulent) is striking, as it displays 156 unique virulence genes not shared with any other pathotype. Interestingly, P0 with a saprophytic lifestyle preference has 24 virulence genes, whereas P1472, a strongly virulent pathotype, has fewer (16), suggesting that it is not only the number of genes which is important but the function of each gene. A comparison within the genus *Colletotrichum*, showed a total of 52 genes, of which 33 were shared by all pathotypes (Figure 8 and Appendix A). Moreover, a few genes were shared by two to three pathotypes and only P1088, P1472, and P2395 were found to have unique genes (Figure 8). The virulence genes shared by all pathotypes included ion and cation channels, plasma membrane ATPase, peroxisomal ATPase, TFs C2H2 zinc-finger, mitogen-activated protein (MAP) kinase (CMK1), fimbrin, nuclease, glucose and hexose transporters, 1,3-β-glucan synthase, 1,6-β-glucan synthase, adenylate cyclase, GTPase activators, chitin synthase (CSHV), polyketide synthase (PKS1), threonine synthase, histidine kinase class IV and pectate lyases among others. We found a laccase gene, the TF Ste12-like transcription factor, a chitin synthase (CHSIII), a MAP kinase (MAF1), a multicopper oxidase, and a hexose transporter only in pathotypes P0, P1088, and P1472. Other genes were unique for one or two pathotypes; for example, P0 and P1088 shared a hexose transporter and the pectate lyase PLB, P1088 and P2395 shared the suppressor of plant defense DN3, whereas P2395 exhibited unique genes, such as the nitrogen starvation-induced glutamine protein, chitin synthase (CHSI), the catalytic subunit of cAMP-dependent protein kinase (CPK1), and the pathogenicity cluster 5 protein d (CLU5d).

### 3.5. CAZymes

Genomic, transcriptomic, and proteomic analyses allowed us to determine that fungi secretes a large number of diverse CAZymes, which are associated with their nutritional strategies and host specificities [21]. CAZymes have a function in carbohydrate synthesis, metabolism, and biotransformation of carbohydrates and are classified into six large families: glycoside hydrolases (GHs), with the carbohydrate binding module (CBM), carbohydrate esterases (CEs), polysaccharide lyases (PLs), glycosyl transferases (GTs), and auxiliary activities (AAs) [94,95]. In this study, the genes encoding CAZymes belonged to 131 families with a high similarity in content for all pathotypes, ranging from 630 to 633 genes (Table 7 and Figure 9). Sixteen families showed a differential number of genes between the pathotypes, and eighteen families were most strongly represented (Figure 10).

Compared to previous studies, the number of CAZymes in *C. lindemuthianum* pathotypes was similar to that reported for *C. fructicola* (668) but higher than in the *C. orbiculare* strain MAFF 240422 (559) and *C. graminicola* (467) [19]. In comparison with other analyses, this number was also higher in *C. lindemuthianum* pathotypes than in the genomes of 102 Colletotrichum species, which ranged from 246 in *C. falcatum* Cf671 to 512 in *C. fructicola* Cg38 [53]. At the intraspecific level, the Mexican *C. lindemuthianum* pathotypes also have more genes encoding CAZymes than the Brazilian strains S83 and S89 (365 and 361, respectively) [53], suggesting an expansion of the gene families in the Mexican pathotypes or otherwise a contraction of these genes in the Brazilian strains (Table 7). Expansion and contraction of CAZymes in the Colletotrichum species genomes has been described previously [20,62,96], specifically in the case of PLs, and this variation has been attributed to fungal lifestyle, host species, and host cell wall pectin content [21,62,97,98,99,100,101].

Our results show that the *C. lindemuthianum* pathotypes P0, P1088, P1472, and P2395 are enriched in hemicellulolytic enzymes, consistent with their high secretion of this activity and their ability to degrade hemicellulose-rich tissues of the host (*P. vulgaris*) and water hyacinth (*Eichhornia crassipes*) [30]. In general, all pathotypes showed a large number of genes encoding GHs (288 to 292), including hemicellulases, cellulases, and debranching enzymes, and a large number of genes encoding AAs (123 to 129), including ligninolytic enzymes (laccases) and lytic polysaccharide monooxygenases (LPMOs). In addition, we found fewer and similar numbers of genes encoding PLs between the pathotypes (37 to 38), including polygalacturonases, pectate, and pectin lyases, among others. Compared with Chen et al. [53], the numbers of GHs (150 to 155), AAs (90 to 93), and PL (27) genes are lower in the Brazilian *C. lindemuthianum* strains S83 and S89 than in the Mexican pathotypes. However, compared with the report by da Silva et al. [102], the number of PL (58) genes in the Brazilian strain S89 was higher than those in Mexican pathotypes. The differences among Mexican and Brazilian populations may be related to their distinct evolutionary histories resulting from their interaction with different bean cultivars and their environmental context.

## 4. Conclusions

Herein, we provide new insights into the nature of intraspecific diversity between *C. lindemuthianum* pathotypes and the origin of their ability to rapidly adapt to changes in the host and environmental conditions. The genome sizes of *C. lindemuthianum* pathotypes and strains are among the largest in the genus due to their high TE contents, which range from 48 to 54.39%. Furthermore, there are differences in the TE content, diversity, and expansion patterns between the P0, P1088, P1472, and P2395 pathotypes and the Brazilian strain S89, suggesting different evolutionary histories. We hypothesize that the high TE and CE contents located in the hypervariable minichromosomes of *C. lindemuthianum* represent the fast-evolving, lifestyle-adapting regions in the dualistic architecture of the two-speed model of evolution in filamentous plant pathogen genomes. Thus, large chromosomes represent the regions of the genome that evolve slowly. In this context, each pathotype showed unique proteins; however, the number of single-copy orthologous genes was identical for all, and the phylogenomic analysis suggested that the pathotypes belong to two genetic groups. The COGs and GO terms of all pathotypes showed the same or similar patterns. In addition, the COGs for carbohydrate transport and metabolism showed the highest number of genes, consistent with the nutritional strategy of the species. TFomes contain the families that are typical in fungal genomes. However, each pathotype exhibited a different total number of TF encoding genes. The most abundant TF encoding genes are those with a fungal-specific TF domain or the Zn2Cys6 domain, including those with a function in virulence. The genomes of all pathotypes contained a similarly high number of peptidases, mainly with a greater abundance of serine peptidases, metallopeptidases, and cysteine peptidases. However, the metallopeptidases were the most diverse. The secretomes showed differences in the number of genes identified across the pathotypes, with the high CE content varying in percentage (from 38 to 40%) in each secretome. The virulence gene contents were high and diverse, with a group of genes being shared for all pathotypes, others being shared by only two or three pathotypes, and genes unique to each pathotype. As expected, the CAZyme gene content was high and diverse, congruent with the nutritional strategy of the species, with hemicellulolytic enzymes being particularly enriched.

## Figures and Tables

**Figure 1 jof-10-00651-f001:**
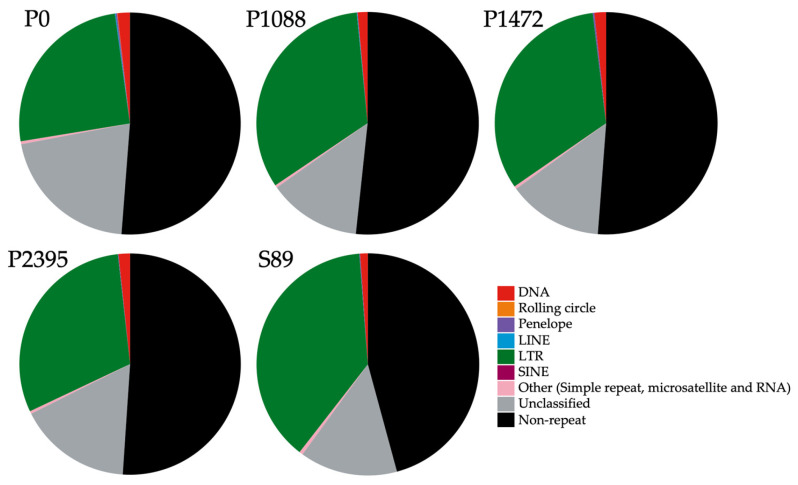
Percentage of repetitive and transposable element families in the genomes of the Mexican pathotypes P0, P1088, P1472, P2395, and the Brazilian strain S89 of *C. lindemuthianum.*

**Figure 2 jof-10-00651-f002:**
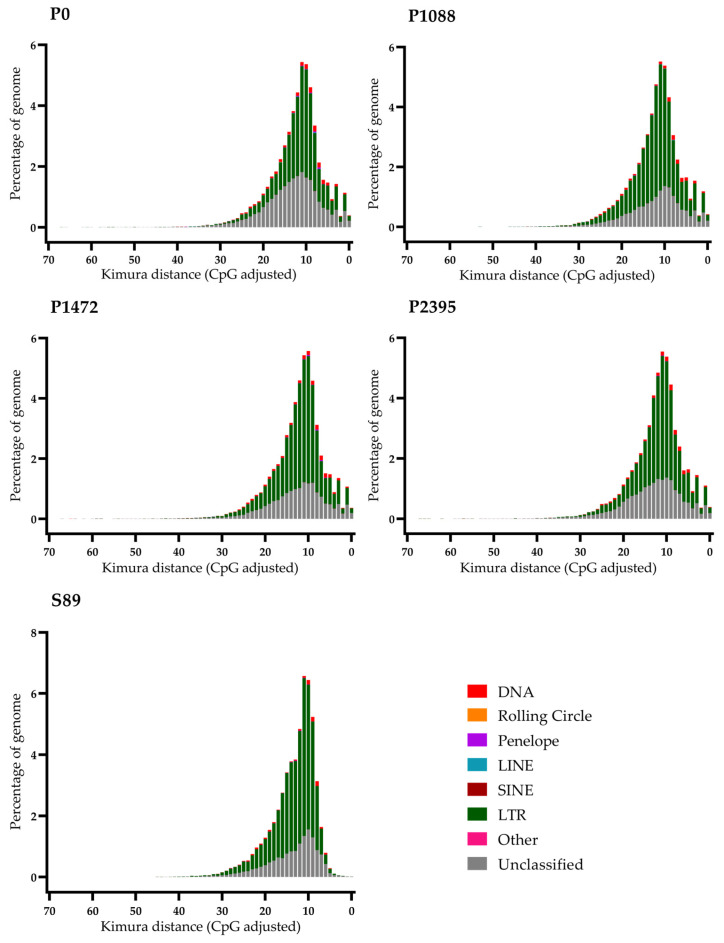
Plots of the copy-number divergence analysis of TE classes, based on Kimura distances for the genomes of Mexican *C. lindemuthianum* pathotypes and strain S89 from Brazil. The percentage of TEs in the genomes is represented on the Y axis and Kimura distance values are plotted on the X axis in which TE copies with recent divergence are close to 0 and copies with previous divergence are close to 60).

**Figure 3 jof-10-00651-f003:**
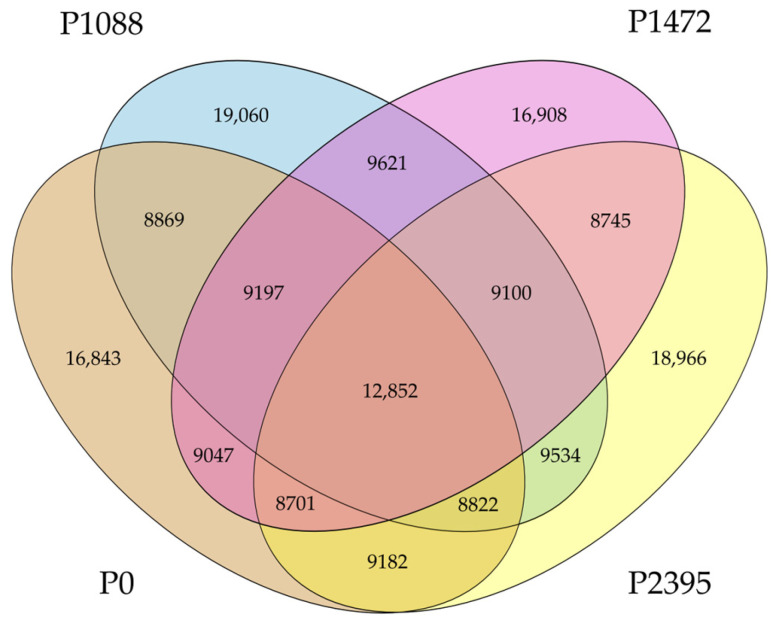
Venn diagram summarizing the annotations shared among *C. lindemuthianum* pathotypes and the unique annotations for each pathotype.

**Figure 4 jof-10-00651-f004:**
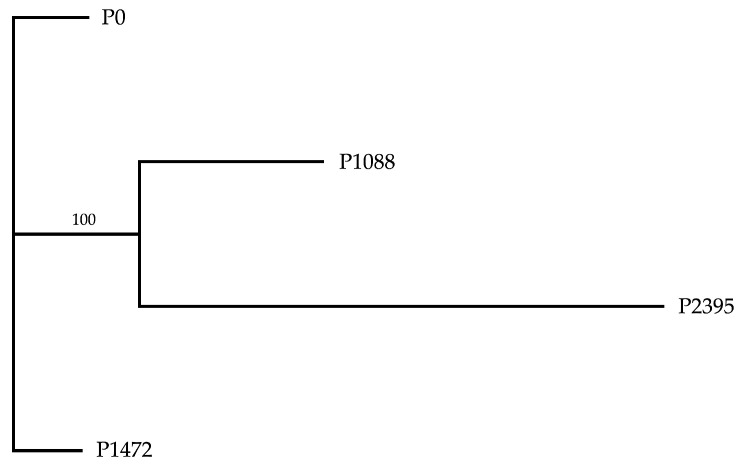
Phylogenomic relationships among the *C. lindemuthianum* pathotypes.

**Figure 5 jof-10-00651-f005:**
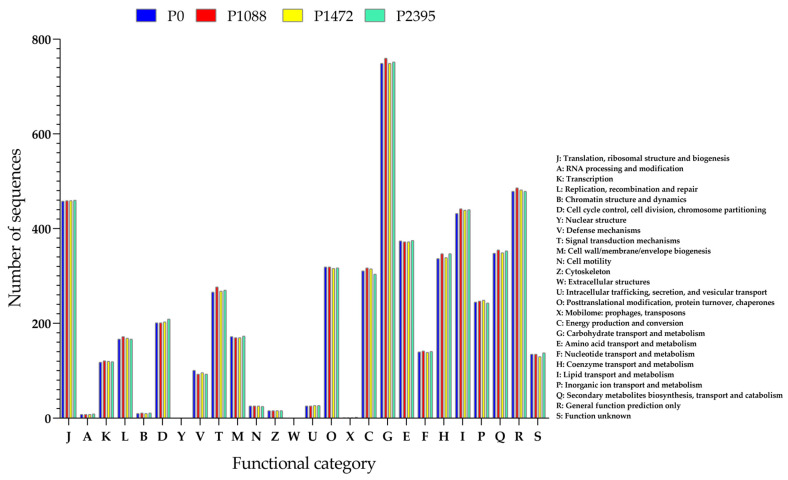
Functional classification of clusters of orthologous genes (COGs) of *C. lindemuthianum* pathotypes.

**Figure 6 jof-10-00651-f006:**
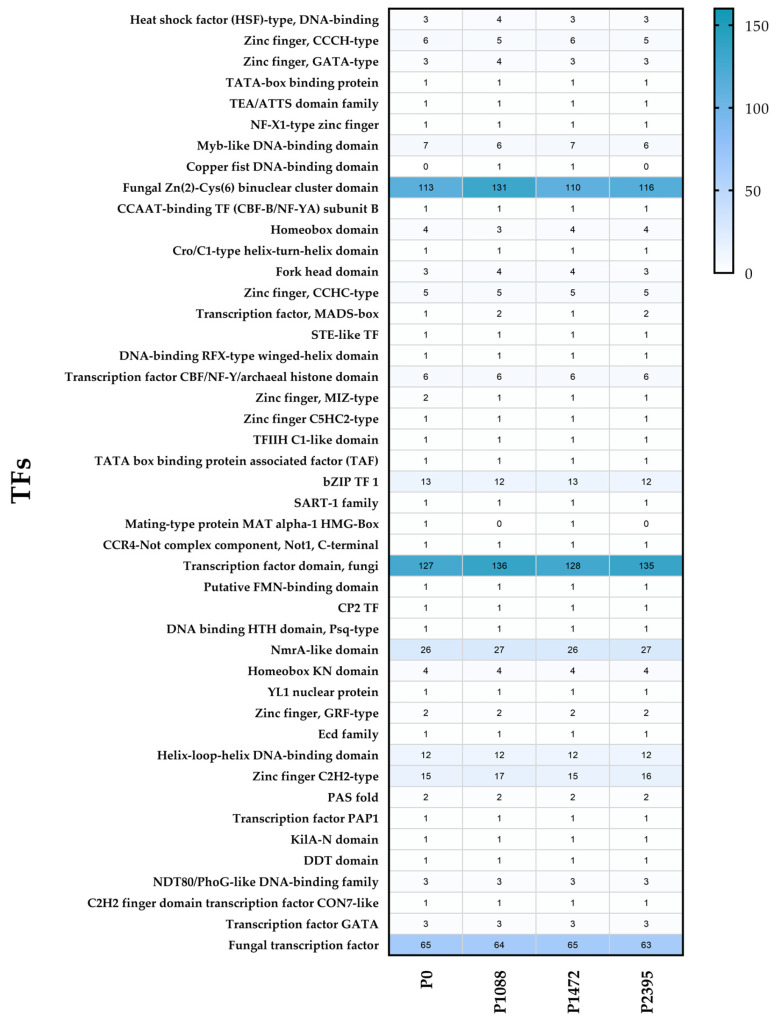
Heatmap representation of numbers of TF genes found in the *C. lindemuthianum* pathotypes.

**Figure 7 jof-10-00651-f007:**
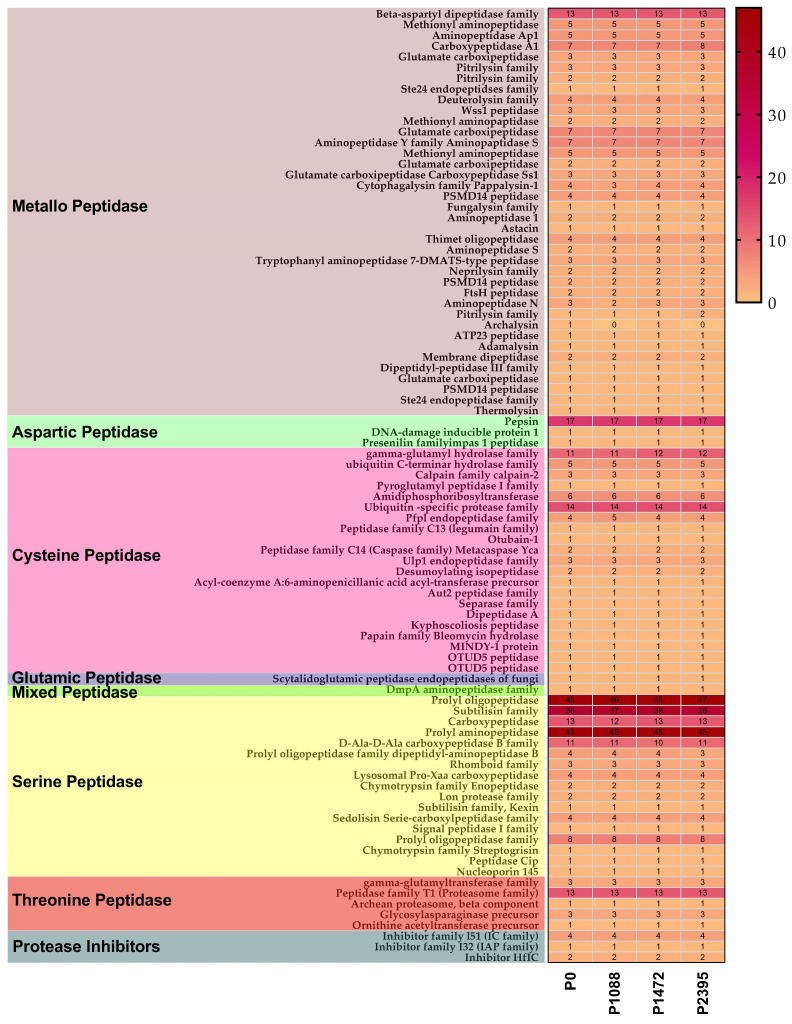
Peptidase families found in *C. lindemuthianum* pathotypes.

**Figure 8 jof-10-00651-f008:**
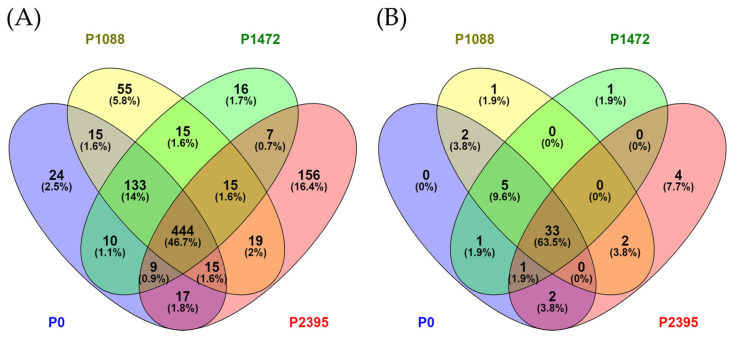
Venn diagrams summarizing the comparison of numbers of virulence factors involved in pathogenicity identified using the plant–pathogen interactions (PHI) database. (**A**) Comparison of the predicted virulence factors of *C. lindemuthianum* pathotypes with the PHI database. (**B**) Comparison of the predicted virulence factors between *C. lindemuthianum* pathotypes and those of the genus *Colletotrichum*.

**Figure 9 jof-10-00651-f009:**
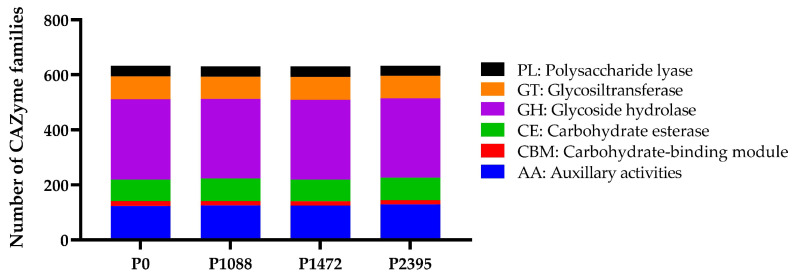
Number of CAZyme families in the *C. lindemuthianum* pathotypes.

**Figure 10 jof-10-00651-f010:**
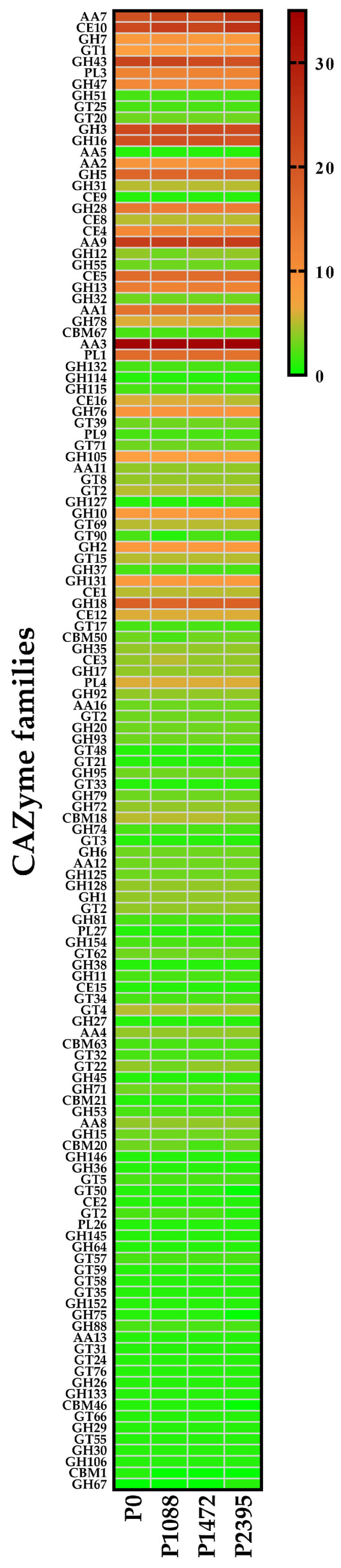
Heatmap representation of the CAZyme families in the *C. lindemuthianum* pathotypes.

**Table 1 jof-10-00651-t001:** Genome assemblies of *C. lindemuthianum* pathotypes P0, P1088, P1472, and P2395, and a comparison with those of *C. lindemuthianum* strains S83 and S89 from Brazil and with those of other *Colletotrichum* species.

Species	Size (Mb)	Scaffolds Number	Scaffolds N50	GC Content (%)	Predicted Genes (Unique)	Assembly
*C. lindemuthianum* P0	98.6	26,178	31,455	38.8	11,871	This study
*C. lindemuthianum* P1088	98.7	29,513	28,426	38.9	12,225	This study
*C. lindemuthianum* P1472	98.1	24,410	33,052	38.8	11,892	This study
*C. lindemuthianum* P2395	101.8	28,615	30,237	38.7	11,859	This study
*C. lindemuthianum* S83.501	97.4	1857	111,300	37.5	11,998	GCA_001693015.2
*C. lindemuthianum* S89 A2 2–3	99.2	1276	158,200	37.5	11,951	GCA_001693025.2
*C. trifolii*	109.7	10,473	35,400	36.5	12,292	GCA_004367215.1
*C. sidae* CBS 518.97	86.8	14,826	39,300	38	12,442	GCA_004367935.1
*C. orbiculare* MAFF 200422	89.7	355	2,078,754	37.5	13,253	GCA_000350065.2
*C. spinosum* CBS 515.97	82.7	10,715	93,600	38.5	12,540	GCA_004366825.1
*C. acutatum* CBS 112980	49.1	389	299,500	52	15,371	GCF_030867785.1
*C. gloeosporioides* Lc1	61.9	128	709,900	51	15,667	GCF_011800055.1
*C. higginsianum* IMI349063	50.7	25	5,200,000	54.5	14,651	GCF_001672515.1
*C. graminicola* M1.001	51.6	653	579,200	49	12,399	GCF_000149035.1
*C. fructicola* Nara gc5	59.5	12	5,500,000	53	17,388	GCA_000319635.2

**Table 2 jof-10-00651-t002:** Transposable elements in the Mexican *C. lindemuthianum* pathotypes and the Brazilian S89 strain.

Family Name	Class	Number				
		P0	P1088	P1472	P2395	S89
DNA	II	3632	3624	3448	3326	827
LINE	I	140	96	0	0	0
LTR	I	36,401	47,786	41,152	45,149	14,814
Other (Simple Repeat, Microsatellite, and RNA)		958	895	916	898	1003
Penelope	I	187	108	240	180	186
Rolling Circle	II	36	0	15	11	8
SINE	I	0	0	18	17	18
Unclassified		31,524	25,673	25,110	28,373	14,133
Total		72,878	78,182	70,899	77,954	30,989

Class I: retrotransposon; Class II: DNA transposons; LINE: long interspersed nuclear elements; LTR: long terminal repeat; SINE: short interspersed nuclear element.

**Table 3 jof-10-00651-t003:** Genome functional annotation of *C. lindemuthianum* pathotypes.

Pathotype	Number of Genes	Number of Proteins	Number of tRNAs	Unique Proteins	Proteins with at Least 1 Ortholog	Single-Copy Orthologous Genes	InterProScan Domains	PFAM Domains
P0	11,871	11,446	425	98	11,348	10,711	12,901	12,528
P1088	12,225	11,797	428	468	11,329	10,711	13,049	12,716
P1472	11,892	11,469	423	101	11,368	10,711	12,913	12,536
P2395	11,859	11,434	425	318	11,116	10,711	12,956	12,696

**Table 4 jof-10-00651-t004:** Peptidase families found in *C. lindemuthianum* pathotypes.

Peptidase Family	P0	P1088	P1472	P2395
Aspartic petidase	19	19	19	19
Cysteine peptidase	62	63	63	63
Glutamic peptidase	1	1	1	1
Metallo peptidase	113	110	113	114
Mixed peptidase	1	1	1	1
Serine peptidase	182	183	183	183
Threonine peptidase	21	21	21	21
Protease inhibitors	7	7	7	7
Total	406	405	408	409

**Table 5 jof-10-00651-t005:** Predicted secretome in *C. lindemuthianum* pathotypes. The number of proteins with SignalP, proteins with transmembrane domain (tmhmm), and secretory proteins (SignalP-excluding-tmhmm) are shown.

Pathotype	All-Predicted-SignalP	All-Predicted-Tmhmm	SignalP-Excluding-Tmhmm
P0	1350	2385	1126
P1088	1440	2460	1197
P1472	1361	2370	1138
P2395	1365	2374	1136

**Table 6 jof-10-00651-t006:** Candidate effectors found in the predicted secretomes of *C. lindemuthianum* pathotypes.

Pathotype	P0	P1088	P1472	P2395
Initial number of proteins	1126	1197	1138	1136
Cytoplasmic effectors	192	212	194	186
Apoplastic effectors	251	263	251	246
Total	443	475	445	432

**Table 7 jof-10-00651-t007:** CAZymes found in the predicted secretomes of the *C. lindemuthianum* pathotypes.

CAZy Family	P0	P1088	P1472	P2395
AA	123	125	124	129
CBM	18	16	16	15
CE	78	82	79	82
GH	292	289	290	288
GT	83	81	83	82
PL	38	38	38	37
Total	632	631	630	633

## Data Availability

The whole-genome sequencing and assembly datasets from this study have been submitted to the NCBI database under the BioProject accession number PRJNA1147206.

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
