# Peer review of "Comparative Genomic Analyses of Colletotrichum lindemuthianum Pathotypes with Different Virulence Levels and Lifestyles"

_jof, 2024, doi:10.3390/jof10090651_

Round 1
Reviewer 1 Report
The manuscript describes the genome sequences, assembly, and functional annotation of three Colletotrichum lindemuthianum pathotypes with varying levels of virulence (P1088, P1472, and P2395) and one non-pathogenic pathotype (P0). Overall, the manuscript is well written, and the research is valuable for advancing our understanding of C. lindemuthianum pathotypes. However, I have a few concerns and suggestions for improvement:
- Abstract
- Line #22: The authors should provide the full terms for COGs and GO instead of using abbreviations. This will make the abstract clearer for readers unfamiliar with the terminology.
- Results and Discussion
- Line #170: Please clarify whether S83 and S89 are strains or races. If possible, provide references to support your explanation.
- Figures
- Figures 2: The clarity and presentation of Figure 2 should be improved for better readability.
- Figures 6 and 7: These figures would benefit from being presented in a landscape format for better visualization and to fit the data more effectively.
By addressing these points, the manuscript will be clearer and more impactful.
By addressing these points, the manuscript will be clearer and more impactful.
The manuscript describes the genome sequences, assembly, and functional annotation of three Colletotrichum lindemuthianum pathotypes with varying levels of virulence (P1088, P1472, and P2395) and one non-pathogenic pathotype (P0). Overall, the manuscript is well written, and the research is valuable for advancing our understanding of C. lindemuthianum pathotypes. However, I have a few concerns and suggestions for improvement:
1. Abstract
- Line #22: The authors should provide the full terms for COGs and GO instead of using abbreviations. This will make the abstract clearer for readers unfamiliar with the terminology.
2. Results and Discussion
- Line #170: Please clarify whether S83 and S89 are strains or races. If possible, provide references to support your explanation.
3. Figures
- Figures 2: The clarity and presentation of Figure 2 should be improved for better readability.
- Figures 6 and 7: These figures would benefit from being presented in a landscape format for better visualization and to fit the data more effectively.
By addressing these points, the manuscript will be clearer and more impactful.

Author Response
Regarding review of our submission, below is the list of responses to your comments. As you will notice, all suggestions were attended and duly replied. We hope that this contribution is now in an acceptable form for publication in the Journal of Fungi. We thank your consideration and peer review which will undoubtedly end in better communication. We will gladly consider further suggestions.
The manuscript describes the genome sequences, assembly, and functional annotation of three Colletotrichum lindemuthianum pathotypes with varying levels of virulence (P1088, P1472, and P2395) and one non-pathogenic pathotype (P0). Overall, the manuscript is well written, and the research is valuable for advancing our understanding of C. lindemuthianum pathotypes. However, I have a few concerns and suggestions for improvement:
- Abstract
Comment:
- Line #22: The authors should provide the full terms for COGs and GO instead of using abbreviations. This will make the abstract clearer for readers unfamiliar with the terminology.
Answer:
It was corrected (Line 22).
- Results and Discussion
Comment:
- Line #170: Please clarify whether S83 and S89 are strains or races. If possible, provide references to support your explanation.
Answer:
We appreciate the observation, S83 and S89 are reported as strains, they do not were reported as confronted with the system of 12 P. vulgaris cultivars. We include this information on lines 177-180.
- Figures
Comment:
- Figures 2: The clarity and presentation of Figure 2 should be improved for better readability.
Answer:
Ok, the Figure 2 was improved (Line 267).
Comment:
- Figures 6 and 7: These figures would benefit from being presented in a landscape format for better visualization and to fit the data more effectively.
Answer:
These Figures are heatmaps, by which the program does not allow horizontal format. Furthermore, if we present the figures horizontally, it is difficult to read the names of the proteins. We consider that the vertical presentation is more appropriate.
By addressing these points, the manuscript will be clearer and more impactful.
Reviewer 2 Report
Specifically, three problems in contents:
1) Page 3 Line 132: this Acc No. was not retrieved on Aug.31 2024.
2) Page 14 Line 462: CEs percentage in P0 should be 39.34%, not 30.34%. Similarly, page 20 line 610: CEs content about 40%, not 30-40%.
3) C. graminicola and C. fructicola are not belonging to C. orbiculare complex. Line556; line 407-408
Minor mistakes such as:
1) Line 608: Divide that sentence before ‘however’
2) Line: add ‘P’ before 2395
3) Line 495: delete as indicated
4) Line492: correct the spelling of ‘eight’
5) Line 446: specifically indicate ‘genes’
6) Line 436: Figure 7, not 8
7) Line 416: correct the organization of this sentence
8) Line 375: delete ‘of’
9) Line 298-299: rephrase this sentence
10) Line 296: add ‘P’ before 2395
11) Line 203-205: rephrase this sentence
12) Line 124, Line 135: better replace this completely repeated content to the end of section 2.2(after Line 121)
13) Line 80-83: rephrase this sentence
14) Line 59: replace ‘of families of genes’ with ‘gene families’
15) Line 39-40: replace ‘where’ with ‘when’
16) Line 27: delete ‘of’ in ‘The of CAZymes’
17) Reference list 15: line 679: show the title in English

Author Response
Regarding review of our submission, below is the list of responses to your comments. As you will notice, all suggestions were attended and duly replied. We hope that this contribution is now in an acceptable form for publication in the Journal of Fungi. We thank your consideration and peer review which will undoubtedly end in better communication. We will gladly consider further suggestions.
The manuscript has undergone English language editing by MDPI in August 2024, Certificate-83994. However, the manuscript was reviewed again by Dr. June Simpson, whose native language is English.
Specifically, three problems in contents:
Comment:
- Page 3 Line 132: this Acc No. was not retrieved on Aug.31 2024.
Answer:
Ok, this is not a problem, access is released once the article is published.
Comment:
- Page 14 Line 462: CEs percentage in P0 should be 39.34%, not 30.34%. Similarly, page 20 line 610: CEs content about 40%, not 30-40%.
Answer:
Ok, it was corrected (Lines 457 and 603).
Comment:
3) C. graminicola and C. fructicola are not belonging to C. orbiculare complex. Line556; line 407-408
Answer:
You are right, we appreciate the observation. It was corrected (Lines 201-203, 403-405 and 551-553).
Minor mistakes such as:
Comment:
- Line 608: Divide that sentence before ‘however’
Answer:
It was corrected (Line 601).
Comment:
2) Line: add ‘P’ before 2395
Answer:
It was corrected (Line 585).
Comment:
3) Line 495: delete as indicated
Answer:
It was deleted (Line 490).
Comment:
4) Line492: correct the spelling of ‘eight’
Answer:
It was corrected (Line 486).
Comment:
5) Line 446: specifically indicate ‘genes’
Answer:
It was corrected (Lines 441-442).
Comment:
6) Line 436: Figure 7, not 8
Answer:
It was corrected (Line 431).
Comment:
7) Line 416: correct the organization of this sentence
Answer:
In was corrected (Lines 413-425).
Comment:
8) Line 375: delete ‘of’
Answer:
In was deleted (Line 371).
Comment:
9) Line 298-299: rephrase this sentence
Answer:
In was corrected (Lines 299-300).
Comment:
10) Line 296: add ‘P’ before 2395
Answer:
In was corrected (Line 298).
Comment:
11) Line 203-205: rephrase this sentence
Answer:
In was corrected (Lines 206-208).
Comment:
12) Line 124, Line 135: better replace this completely repeated content to the end of section 2.2 (after Line 121)
Answer:
In was corrected (Lines 124-125).
Comment:
13) Line 80-83: rephrase this sentence
Answer:
In was corrected (Lines 80-83).
Comment:
14) Line 59: replace ‘of families of genes’ with ‘gene families’
Answer:
In was corrected (Line 59).
Comment:
15) Line 39-40: replace ‘where’ with ‘when’
Answer:
In was corrected (Lines 39-40).
Comment:
16) Line 27: delete ‘of’ in ‘The of CAZymes’
Answer:
In was corrected (Line 27).
Comment:
17) Reference list 15: line 679: show the title in English
Answer:
In all literature where this reference is cited, it is in Spanish, as it was published. As an example, see in Nunes et al. (2021) [13]. We consider that there is no problem in it being in Spanish.

Round 2
Reviewer 2 Report
This work provided knowledge in the genomic diversity of Colletotrichum lindemuthianum, which improved our understanding of the
plasticity and virulence of this important pathogen species.
line 207: correct 'similar lower'
line 208: correct 'higher similar'
line 313: correct 'by meant of'
line 500: correct '444 of out of'
Author Response
Regarding review of our submission, below is the list of responses to your comments. As you will notice, all suggestions were attended and duly replied. We hope that this contribution is now in an acceptable form for publication in the Journal of Fungi. We thank your consideration and peer review which will undoubtedly end in better communication. We will gladly consider further suggestions.
Major Comments
This work provided knowledge in the genomic diversity of Colletotrichum lindemuthianum, which improved our understanding of the plasticity and virulence of this important pathogen species.
Detail comments
Comments:
line 207: correct 'similar lower'
line 208: correct 'higher similar'
Answer:
We rewrite the sentence (Lines 206-207).
Comment:
line 313: correct 'by meant of'
Answer:
It was corrected (Line 311).
Comment:
line 500: correct '444 of out of'
Answer:
It was corrected (Line 498).